# Indication of frailty transitions on 2-year adverse health outcomes among older Chinese inpatients: Insight from a multicenter prospective cohort study

**Miao Yu[1‡], Jiaqi Ding[1], Xinjuan Wu[2], Xianxiu Wen[3], Jingfen Jin[4], Hui Wang[5], Dongmei Lv[6], Shengxiu Zhao[7], Jing Jiao[2]\*, Tao Xu[1]\***

1 Department of Epidemiology and Statistics, Institute of Basic Medical Sciences, Chinese Academy of Medical Sciences & School of Basic Medicine, Peking Union Medical College, Dongcheng District, Beijing, China, 2 Department of Nursing, Chinese Academy of Medical Sciences, Peking Union Medical College, Peking Union Medical College Hospital (Dongdan campus), Beijing, China, 3 Department of Nursing, Sichuan Provincial People's Hospital, Chengdu, China, 4 Department of Nursing, The Second Affiliated Hospital Zhejiang University School of Medicine, Hangzhou, China, 5 Department of Nursing, Tongji Hospital, Tongji Medical College, Huazhong University of Science and Technology, Wuhan, Chin, 6 Department of Nursing, The Second Affiliated Hospital of Harbin Medical University, Harbin, China, 7 Department of Nursing, Qinghai Provincial People's Hospital, Xining, China

‡ MY is senior author on this work.
* xutaosd@126.com (TX); jiaojing.2006@aliyun.com (JJ)

## Abstract

## Introduction

Frailty is thought to be associated with an increased risk of adverse health outcomes such as death and falls, but comparatively little is known about the impact of frailty transitions on the adverse health outcomes. Moreover, owing to insufficient sample size or a single-center study design, previous studies have not been sufficiently representative of elderly inpatients in China. This study aimed to provide estimates at the population level of the association between frailty transitions and adverse outcomes among elderly inpatients following discharge.

## Methods

This was a large-scale multicenter cohort study conducted from October 2018 to February 2021. The FRAIL scale was used to estimate frailty status. Frailty transitions were derived by considering frailty status at baseline and the 3-month follow-up, which encompassed five patterns: persistent non-frailty, persistent pre-frailty, persistent frailty, improvement in frailty, and worsening of frailty. The outcome variables included mortality, falls, hospital readmissions, and Health-Related Quality of Life (HRQoL). Cox proportional hazard regression, generalized linear models and linear regression was used to examine the association between frailty transitions and adverse health outcomes.

**Data Availability Statement:** The data contains potentially identifying or sensitive patient information, and the Research Ethics Committee

has imposed ethical restrictions. The data cannot be shared publicly as it comes from an elderly population in China and is protected by ethical guidelines that prohibit data sharing. The study adheres to the ethical principles of voluntariness, confidentiality, fairness, and ensuring benefits without harm, as reviewed and approved by the Ethics Review Committee of Peking Union Medical College Hospital, Chinese Academy of Medical Sciences. Expedited Review Content: The study aims to describe the prevalence of frailty among hospitalized elderly patients in China, analyze its risk factors, explore the relationship between frailty and health outcomes, and assess the applicability of various frailty assessment tools. Hospitals will be selected based on China's geographical distribution and economic levels, with one tertiary hospital chosen from each province within six administrative regions. Both internal medicine and surgical patients from these hospitals will be included in the survey. A total of 8,400 cases will be collected, with the survey covering general patient information, disease-related data, frailty assessment, depression status, and activities of daily living. The study adheres to the ethical principles of voluntariness, confidentiality, fairness, and ensuring benefits without harm. The relevant ethical approval will be uploaded as a file under the "other-ethics statement".

**Funding:** This study received financial support by the National High Level Hospital Clinical Research Funding (2022-PUMCH-C-058), Capital's Funds for Health Improvement and Research(2024-2G-4211), the CAMS Innovation Fund for Medical Sciences (2018-I2M-AI-009) and the Special Research Fund for Central Universities, Peking Union Medical College (2018PT33001). The funders had no role in study design, data collection and analysis, decision to publish, or preparation of the manuscript.

**Competing interests:** NO.

## Results

A total of 8,256 patients were included in the study, 40.70% of study participants were non-frail, 43.04% were pre-frail, and 16.27% were frail. Compared with patients who persistently non-frail patients, those who frailty improvement, persistent pre-frailty, worsening frailty, and persistent frailty showcased escalated risks of mortality within 2 years after enrollment [$HR$ (95% CI): 1.32 (1.06–1.64)], 1.71 (1.37–2.13), 2.43 (1.95–3.02), and 2.44 (1.81–3.29), respectively. These groups also faced elevated hazards of 2-year falls [$OR$(95% CI): 1.586 (1.13–2.23), 2.21(1.55–3.15), 1.94(1.33–2.82), 2.71(1.59–4.62)] and re-hospitalization risk within 2 years[$OR$(95% CI): 1.33(1.13–1.56), 1.56(1.32–1.86), 1.53(1.28–1.83), 2.29(1.74–3.01). The number of falls increased by 0.76 over 2 years in frailty-worsened patients and 0.81 in persistently pre-frail patients. The total days of rehospitalization increased by 0.35 over 2 years in frailty-improved patients, by 0.61 in frailty-worsened patients, by 0.66 in elderly in persistently pre-frail patients and by 0.80 in persistently frail patients. Moreover, patients exhibiting frailty-improved [-1.23 (95% CI: -2.12 to -0.35)], persistently pre-frail [-4.95 (95% CI: -5.96 to -3.94)], frailty-worsened [-3.67 (95% CI: -4.71 to -2.62)], and persistently frail [-9.76 (95% CI: -11.60 to -7.93)] displayed inverse correlations with the regression coefficients of HRQoL.

## Discussion

Frailty-improved, worsened, persistently pre-frail, and frail inpatients face higher risks of mortality, falls, rehospitalization, reduced HRQoL than consistently non-frail inpatients. Screening for frailty among elderly inpatients can identify individuals at increased risk of adverse health outcomes.

## 1 Background

China's population is already aging, and it will continue to age rapidly in the coming decades. In 2021, 14.2% of China's population was aged 65 and over, signifying that the proportion of elderly individuals aged 65 and above in China has doubled within 21 years, increasing from 7% to 14% [1]. However, the exclusive reliance on chronological age was found to be insufficient for predicting the health outcomes of elderly individuals due to the significant variability in physical conditions that were observed among individuals of the same age group. In contrast, frailty was recognized as a valuable metric for assessing an individual's biological age, providing a more precise indicator of the overall health and aging status of older adults [2]. With the rapid progression of population aging, frailty has been increasingly recognized for its significant impact on healthcare systems worldwide [3]. Frailty was regarded as a clinical syndrome characterized by a reduced ability to withstand stress due to diminished physical reserves in the elderly population. Even minor external stimuli were found to be capable of causing disproportionate harm to the health of frail individuals [4]. Studies have shown that the prevalence of frailty ranges from 2.6%–19.6% in the community-dwelling elderly population [5–7] and 25.2%–48.8% in the hospitalized elderly population [8–12]. To promote "healthy aging", frailty has emerged as a critical public health problem to be addressed [13].

Due to accentuated physiological decline in several systems such as neuromuscular and immune systems, frail elderly people are at a higher risk of adverse outcomes such as falls,

hospitalizations, and death [14, 15], as well as a 1.54-fold increase in health and long-term care needs [4], in comparison with their non-frail counterparts. Moreover, frailty is a dynamic process, with deterioration occurring more commonly than improvement [16]. However, in most studies, frailty at baseline in the elderly population was solely employed to evaluate its relationship with intermediate and long-term adverse health outcomes [17, 18]. This approach overlooked the potential fluctuations in frailty status over time.

Previous research has generally overlooked the segment of elderly adults receiving hospital care, despite their tendency to exhibit suboptimal physical health, an elevated prevalence of frailty, and heightened susceptibility to adverse consequences compared to their counterparts residing in the community [19]. Consequently, the implementation of frailty assessment protocols within the hospital setting holds promise for the identification of individuals at an elevated peril of adverse health events. This, in turn, facilitates the formulation of tailored clinical care strategies aimed at enhancing patient prognostication and mitigating the likelihood of adverse outcomes. Hence, it is imperative to conduct comprehensive investigations into the frailty status of elderly patients admitted to hospitals.

In China, prior investigations on frailty have exhibited limitations such as inadequate representation and limited scale. To address these shortcomings, a nationwide multicenter cohort study, based in hospitals, was conducted. This comprehensive study aimed to explore the relationship between transitions in frailty status and adverse health outcomes in a national sample elderly Chinese inpatient population. This study aims to use the Health-Related Quality of Life (HRQoL) metric to assess patients' health and living conditions. HRQoL is defined by the World Health Organization as "an individual's perception of their position in life, in the context of the culture and value systems in which they live, and in relation to their goals, expectations, standards, and concerns" [20]. The outcomes under scrutiny included mortality rates, instances of falls, frequency of falls, hospital readmissions, length of stay during readmissions, and the deterioration in HRQoL.

## 2. Materials and methods

### 2.1 Study population

Through a collaborative effort involving Peking Union Medical College Hospital and five distinguished hospitals located in different provinces and cities across China, the data for this study were acquired. This extensive multicenter cohort study, conducted between October 2018 and February 2021, aimed to evaluate the mental and physical well-being of elderly hospitalized patients in China. Details regarding the study's design and data collection were previously published [21–23]. A total of 9996 inpatients from the surgical and internal medicine departments of six hospitals were enrolled, meeting the following inclusion criteria: age 65 or older, voluntary participation with informed consent, and ability to communicate effectively or have caregivers provide accurate information. Exclusion criteria included: Patients with persistent consciousness or communication disorders, and whose caregivers could not provide accurate information; Those lost to follow-up during the 2-year period. Before starting the study, one or two nurses from each department were selected as investigators and received standardized training on study procedures, data collection, communication, and the EDC system, with a focus on data accuracy and quality control, along with periodic refresher sessions. Patients underwent telephone follow-ups at one month, three months, one year, and two years after enrollment. This study received approval from the ethics committee of Peking Union Medical College Hospital (S-K540) (Fig 1).

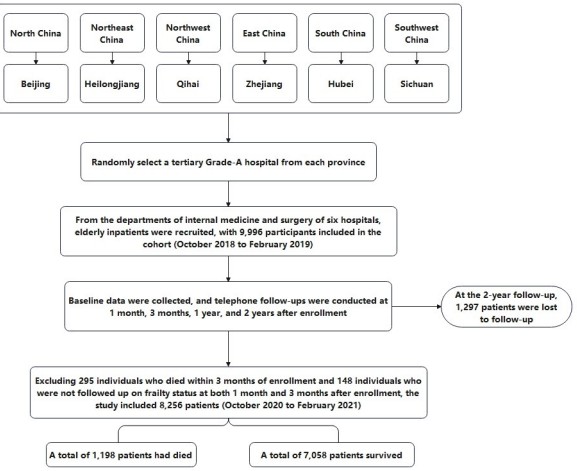

**Fig 1. Flowchart of participant inclusion and exclusion.**

## 2.2 Study measures

### 2.2.1 FRAIL (Fatigue, resistance, ambulation, illness, and loss of weight) scale

Frailty was assessed using the FRAIL scale, consisting of five questions with each item assigned a score of 0 or 1 [24]. A total score of 0 on the five components indicated robustness or non-frailty, while a score of 1–2 signified pre-frailty, and 3–5 indicated frailty. The FRAIL scale demonstrated robust reliability and validity in a cross-sectional study involving Chinese community-dwelling older adults [25].

### 2.2.2 Outcomes

The primary outcome of this study was mortality within the 2 years following enrollment, measured as the time to the event. Secondary outcomes included falls, the number of falls, hospital readmissions, readmission length of stay, and HRQoL. The number of falls referred to the total instances of falls experienced by elderly patients within two years of enrollment. Hospital readmission was defined as any subsequent hospital admission within 2 years after the initial hospitalization discharge. Readmission length of stay was computed by summing the length of stay during each readmission [26–28]. We assessed HRQoL utilizing the EuroQol Visual Analogue Scale (EQ-VAS), a tool previously demonstrated to exhibit acceptable feasibility, acceptability, and reliability within the Chinese population [19, 29]. Participants assessed their overall health status by marking lines on the scale, where a score of 100 denoted "perceived best health condition," and a score of 0 represented "perceived worst health condition."

### 2.2.3 Covariates

We gathered a total of 25 variables, encompassing gender, age, ethnicity, educational level, marital status, Body Mass Index (BMI), history of surgery (defined as having undergone surgery at any point in life), smoking and drinking habits, falls within the past year, bedridden status for $\geq$ 4 weeks, polypharmacy, nutritional status, presence of depression, cognitive

function, handgrip strength, vision and hearing capabilities, sleep patterns, urination and defecation patterns, tumor presence, Activities of Daily Living (ADL), and Instrumental Activities of Daily Living (IADL). Baseline data were acquired through questionnaire interviews, physical examinations, clinical records, and clinical assessments. BMI was categorized into four groups: underweight ($< 18.5$ kg/m$^2$), normal weight (18.5 to $< 24$ kg/m$^2$), overweight (24 to $< 28$ kg/m$^2$), and obese ($\geq 28.0$ kg/m$^2$) [30].

The Geriatric Depression Scale-15 was used to evaluate depression [31]. A score of 5 or higher on this scale indicates the presence of depression, with higher scores indicative of greater depression severity. Cognitive function was assessed using the Mini Cognitive Scale [32]. Patients were categorized as having cognitive impairment if their score was 0, or if their score ranged from 1 to 2 and they demonstrated poor performance on the clock-drawing test. Conversely, patients were classified as having no cognitive impairment if their score was 3 or if their score ranged from 1 to 2 and they accurately drew the clock. The Mini-Nutritional Assessment Short Form was utilized to evaluate the nutritional status of older adults [33]. The scale comprises a total score of 14 points: scores ranging from 0 to 7 points indicate malnutrition, scores between 8 and 11 points suggest a risk of malnutrition, and scores of 12 to 14 points imply a normal nutritional status. To evaluate the ADL ability of frail older patients, the Barthel Index was employed [34]. Based on their scale scores, patients were categorized into three groups: good (61–100 points), moderate (41–60 points), and poor ($\leq$40 points). To assess the functional ability of older adults in daily living tasks, the IADL scale was employed [35]. The total score on the IADL scale ranges from 0 to 8 points, with a higher score signifying superior performance in IADL.

## 2.3 Statistical analysis

Frailty transitions were derived by considering frailty status at baseline and the 3-month follow-up from enrollment. During this period, frailty status was categorized as frailty (FRAIL scale score 3 to 5), pre-frailty (FRAIL scale score 1 to 2), or non-frailty (FRAIL scale score 0). In cases where frailty status was missing at the 3-month follow-up, the status at the 1-month follow-up was utilized to fill the gap, employing the Last Observation Carried Forward (LOCF) method. Frailty transitions encompassed five patterns: persistent non-frailty, persistent pre-frailty, persistent frailty, improvement in frailty (including three groups: transitioning from frail to non-frail, frail to pre-frail, and pre-frail to non-frail), and worsening of frailty (including three groups: transitioning from non-frail to frail, pre-frail to frail, and non-frail to pre-frail).

In the analysis of baseline characteristics, descriptive statistics were employed, presenting variables as means (standard deviations) or frequencies (percentages). Kruskal–Wallis tests were utilized for between-group comparisons. Kaplan–Meier survival curves were employed to illustrate cumulative survival among participants categorized into the five groups based on their frailty transition patterns within 2 years following hospitalization. Survival curves were compared across frailty transition patterns using the log-rank test. Hazard ratios (*HRs*) with corresponding 95% confidence intervals (*CIs*) were calculated using Cox regression to assess the association between frailty transitions and mortality.

Odds ratios (*ORs*) and 95% *CIs* were determined using logistic regression to examine the relationship between frailty transitions and incidents of falls and rehospitalization. Negative binomial regression within generalized linear models was applied to calculate regression coefficients and 95% *CIs*, assessing the association between frailty transitions and the number of falls and the total duration of rehospitalization. Linear regression was used to evaluate the association between frailty transitions and the reduction in HRQoL. All statistical tests were two-

sided, with statistical significance set at $p < 0.05$. Data analyses were performed using SAS Enterprise Guide Version 9.4 (SAS Institute Inc., Cary, NC, USA).

## 3. Results

### 3.1 Baseline information

Overall, a total of 9,996 older patients in six hospitals across China met the inclusion criteria and were included in the study. After 2-year follow-up, 1,297 patients were lost to follow-up. Excluding 295 patients who died within three months of enrollment, as well as 148 patients for whom frailty status data were not recorded during the 1-month and 3-month follow-up visits, a total of 8256 patients completed the baseline survey and remained in the cohort. According to the FRAIL scale, 40.70% of study participants were non-frail, 43.04% were pre-frail, and 16.27% were frail. S1 Table shows the baseline information of patients. Among all patients, 42.47% were women and 57.53% were men. Participants aged 65–74 years and 75–84 years accounted for 71.22% and 25.63%, respectively; 94.14% of participants were Han nationality and 88.85% were married. In total, 40.56% of patients had a middle school degree and 28.94% had a primary school degree. Participants with normal weight and overweight status accounted for 48.36% and 35.11%, respectively, and 14.22% had fallen within the past year. Patients with a risk of malnutrition and those with malnutrition accounted for 33.83% and 9.73%, respectively; 49.04% of patients had low grip strength, 15.49% had depression, and 19.81% had cognitive impairment. Sleep abnormalities and urinary disorders were found in 42.83% and 13.12% of participants, respectively. Patients with tumor accounted for 25.16% of the total. There were significant differences in age, sex, marital status, ethnicity, education level, BMI, smoking status, drinking status, bedridden status, polypharmacy, falls, and sleep among patients with non-frail, pre-frail, and frail status. At three months after enrollment, 30.09% of the subjects remained non-frail, 18.20% remained pre-frail and 6.25% remained frail, while others' frailty status has transitioned, with 15.90% of them worsening and 29.55% improving.

### 3.2 Comparison of Kaplan–Meier survival curves

The Kaplan–Meier method was used to estimate the survival rate in the four groups of participants with different frailty transition patterns. As shown in Fig 2, the survival curve of the persistently non-frail group was the highest, followed by the frailty-improved group, the

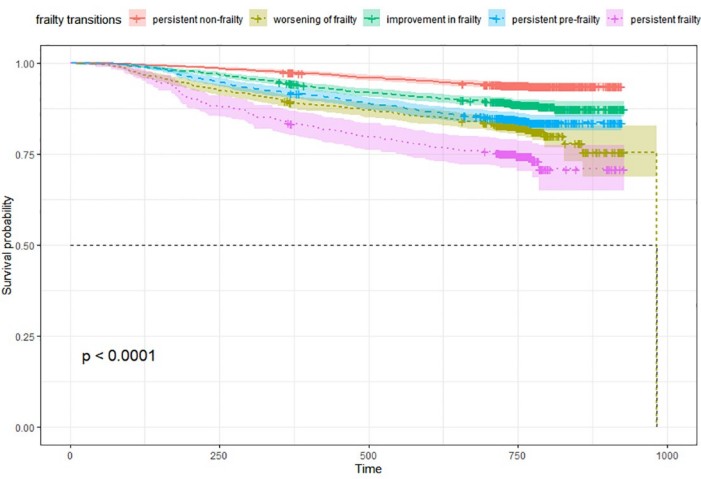

**Fig 2. Survival curves in different frailty transition status.**

**Table 1. Cox regression analysis of the relationship between frailty transitions and 2-year death.**

| Indicators | Unadjusted | | | | | Adjusted* | | | | |
|---|---|---|---|---|---|---|---|---|---|---|
| | Remaining non-frail | Remaining pre-frail | Remaining frail | Improvement | Deterioration | Remaining non-frail | Remaining pre-frail | Remaining frail | Improvement | Deterioration |
| *HR* | 1(Ref.) | 2.69 | 4.80 | 1.87 | 3.15 | 1(Ref.) | 1.71 | 2.44 | 1.32 | 2.43 |
| 95%*CI* | 1(Ref.) | 2.19, 3.31 | 3.78, 6.10 | 1.53, 2.28 | 2.56, 3.87 | 1(Ref.) | 1.37 2.13 | 1.81, 3.29 | 1.06, 1.64 | 1.95, 3.02 |
| *P* | - | <0.01 | <0.01 | <0.01 | <0.01 | - | <0.01 | <0.01 | <0.01 | <0.01 |

Adjusted: gender, age, ethnicity, educational level, marital status, BMI, surgery, smoking status, drinking status, falls in the past year, bedridden for $\geq$ 4 weeks, multidrug, nutritional status, depression, cognition, handgrip strength, vision, hearing, sleep, urination, defecation, tumor, ADL, IADL

persistently pre-frail group, the frailty-worsened group and the persistently frail group's survival curve was the lowest. Significant differences were found among the survival curves of the four groups after performing the log-rank test. This indicated that as the frailty conditions of elderly patients worsened, their survival rates also decreased.

### 3.3 Cox regression analysis of the relationship between frailty transitions and 2-year death

Table 1 shows that when we adjusted confounders such as age, gender and depression status, compared with persistently non-frail patients, the *HR* for death within 2 years after enrollment was 1.32 (95% confidence interval [CI]: 1.06–1.64) in frailty-improved patients, 1.71 (95% CI: 1.37–2.13) in persistently pre-frail patients, 2.43 (95% CI: 1.95–3.02) in frailty-worsened patients, and 2.44 (95% CI: 1.81–3.29) in persistently frail patients. These results indicated that the risk of death within 2 years was 1.32 times higher in frailty-improved patients, 1.71 times higher in persistently pre-frail patients, 2.426 times higher in frailty-worsened patients, and 2.443 times higher in persistently frail patients, compared with the persistently non-frail patients. It can be observed that the patient's risk of death gradually increases as the frailty condition deteriorates.

### 3.4 GLM analysis of the relationship between frailty transitions and falls and the number of falls within 2 years

Table 2 shows that when we adjusted confounders such as age, gender and depression status, the *OR* for falls within 2 years after enrollment was 1.59 (95% CI: 1.13–2.23) in frailty-improved patients, 1.94 (95% CI: 1.33–2.82) in frailty-worsened patients, 2.21 (95% CI: 1.55–3.15) in persistently pre-frail patients, and 2.71 (95% CI: 1.59–4.62) in persistently frail patients. These results indicated that the risk of falls within 2 years was 1.59 times higher in frailty-improved patients, 1.94 times higher in frailty-worsened patients, 2.21 times higher in persistently pre-frail patients, and 2.71 times higher in persistently frail patients, compared with the persistently non-frail patients.

In the model with adjustment for confounders, the regression coefficient for the number of falls within 2 years after enrollment was 0.76 (95% CI: 0.31–1.22) for frailty-worsened patients and 0.81 (95% CI: 0.37–1.25) for persistently pre-frail patients. Compared with the persistently non-frail patients, the number of falls increased by 0.76 over 2 years in frailty-worsened patients and 0.81 in persistently pre-frail patients.

**Table 2. GLM analysis of the relationship between frailty transitions and 2-year falls and the number of falls within 2 years.**

| Indicators | Unadjusted | | | | | Adjusted* | | | | |
|---|---|---|---|---|---|---|---|---|---|---|
| | Persistent non-frailty | Persistent pre-frailty | Persistent frailty | Improvement in frailty | Worsening of frailty | Persistent non-frailty | Persistent pre-frailty | Persistent frailty | Improvement in frailty | Worsening of frailty |
| Falls* | | | | | | | | | | |
| OR | 1(Ref.) | 2.57 | 3.16 | 1.71 | 2.38 | 1(Ref.) | 2.21 | 2.71 | 1.59 | 1.94 |
| 95%CI | 1(Ref.) | 1.86, 3.56 | 2.03, 4.91 | 1.25, 2.33 | 1.69, 3.36 | 1(Ref.) | 1.55, 3.15 | 1.59, 4.62 | 1.13, 2.23 | 1.33, 2.82 |
| P | - | <0.01 | <0.01 | <0.01 | <0.01 | - | <0.01 | <0.01 | 0.01 | <0.01 |
| The number of falls* | | | | | | | | | | |
| Coefficient | 0(Ref.) | 1.01 | 0.97 | 0.40 | 1.11 | 0(Ref.) | 0.81 | 0.48 | 0.08 | 0.76 |
| 95%CI | 0(Ref.) | 0.60, 1.43 | 0.34, 1.60 | 0.01, 0.79 | 0.68, 1.54 | 0(Ref.) | 0.37, 1.25 | -0.27, 1.24 | -0.35, 0.51 | 0.31, 1.22 |
| P | - | <0.01 | <0.01 | 0.04 | <0.01 | - | <0.01 | 0.21 | 0.71 | 0.01 |

Adjusted: gender, age, ethnicity, educational level, marital status, BMI, surgery, smoking status, drinking status, falls in the past year, bedridden for $\geq$ 4 weeks, multidrug, nutritional status, depression, cognition, handgrip strength, vision, hearing, sleep, urination, defecation, tumor, ADL, IADL

Falls: Using the logistic regression in generalized linear model

The number of falls: Using the negative binomial regression in the generalized linear model

### 3.5 GLM analysis of the relationship between frailty transitions and rehospitalization and total days of rehospitalization within 2 years

Table 3 shows that when the above confounders were adjusted, the *OR* for rehospitalization within 2 years after enrollment was 1.33 (95% *CI*: 1.13–1.56) in frailty-improved patients, 1.53 (95% *CI*: 1.28–1.83) in frailty-worsened patients, 1.56 (95% *CI*: 1.32–1.86) in persistently pre-frail patients, and 2.29 (95% *CI*: 1.74–3.01) in persistently frail patients. These results indicated that the risk of rehospitalization within 2 years was 1.33 times higher in frailty-improved patients, 1.53 times higher in frailty-worsened patients, 1.56 times higher in persistently pre-frail patients, and 2.29 times higher in persistently frail patients, compared with the persistently non-frail patient.

**Table 3. GLM analysis of the relationship between frailty transitions and 2-year rehospitalization and total days of rehospitalization within 2 years.**

| Indicators | Unadjusted | | | | | Adjusted* | | | | |
|---|---|---|---|---|---|---|---|---|---|---|
| | Persistent non-frailty | Persistent pre-frailty | Persistent frailty | Improvement in frailty | Worsening of frailty | Persistent non-frailty | Persistent pre-frailty | Persistent frailty | Improvement in frailty | Worsening of frailty |
| Rehospitalization* | | | | | | | | | | |
| OR | 1(Ref.) | 2.10 | 3.29 | 1.55 | 1.89 | 1(Ref.) | 1.56 | 2.29 | 1.33 | 1.53 |
| 95%CI | 1(Ref.) | 1.80, 2.44 | 2.65, 4.09 | 1.34 1.78 | 1.61, 2.22 | 1(Ref.) | 1.32, 1.86 | 1.74, 3.01 | 1.13, 1.56 | 1.28, 1.83 |
| P | - | <0.01 | <0.01 | <0.01 | <0.01 | - | <0.01 | <0.01 | <0.01 | <0.01 |
| The total days of rehospitalization* | | | | | | | | | | |
| Coefficient | 0(Ref.) | 0.86 | 1.57 | 0.53 | 0.86 | 0(Ref.) | 0.67 | 0.80 | 0.35 | 0.61 |
| 95%CI | 0(Ref.) | 0.55, 1.17 | 1.08, 2.06 | 0.27, 0.79 | 0.54, 1.19 | 0(Ref.) | 0.33, 0.99 | 0.21, 1.40 | 0.05, 0.65 | 0.28, 0.95 |
| P | - | <0.01 | <0.01 | <0.01 | <0.01 | - | <0.01 | 0.01 | 0.02 | <0.01 |

Adjusted: gender, age, ethnicity, educational level, marital status, BMI, surgery, smoking status, drinking status, falls in the past year, bedridden for $\geq$ 4 weeks, multidrug, nutritional status, depression, cognition, handgrip strength, vision, hearing, sleep, urination, defecation, tumor, ADL, IADL

Rehospitalization: Using the logistic regression in generalized linear model

The total days of rehospitalization: Using the negative binomial regression in the generalized linear model

**Table 4. Linear regression analysis of the relationship between frailty transitions and 2-year HRQoL.**

| Indicators | Unadjusted | | | | | Adjusted* | | | | |
|---|---|---|---|---|---|---|---|---|---|---|
| | Persistent non-frailty | Persistent pre-frailty | Persistent frailty | Improvement in frailty | Worsening of frailty | Persistent non-frailty | Persistent pre-frailty | Persistent frailty | Improvement in frailty | Worsening of frailty |
| Coefficient | 0(Ref.) | -7.24 | -16.07 | -3.86 | -5.11 | 0(Ref.) | -4.95 | -9.76 | -1.23 | -3.67 |
| 95%*CI* | 0(Ref.) | -8.22, -6.26 | -17.67, -14.47 | -4.70 -3.03 | -6.15, -4.07 | 0(Ref.) | -5.96, -3.94 | -11.60, -7.93 | -2.12, -0.35 | -4.71, -2.62 |
| *P* | - | <0.01 | <001 | <0.01 | <0.01 | - | <0.01 | <0.01 | 0.01 | <0.01 |

Adjusted: gender, age, ethnicity, educational level, marital status, BMI, surgery, smoking status, drinking status, falls in the past year, bedridden for $\geq$ 4 weeks, multidrug, nutritional status, depression, cognition, handgrip strength, vision, hearing, sleep, urination, defecation, tumor, ADL, IADL

In the model with confounders adjusted, the regression coefficient of the total days of rehospitalization within 2 years after enrollment was 0.35 (95% *CI*: 0.05–0.647) for frailty-improved patients, 0.61 (95% *CI*: 0.28–0.95) for frailty-worsened patients, 0.66 (95% *CI*: 0.33–0.99) for persistently pre-frail patients and 0.80 (95% *CI*: 0.21–1.40) for persistently frail patients. Compared with the persistently non-frail patients, the total days of rehospitalization increased by 0.35 over 2 years in frailty-improved patients, by 0.61 in frailty-worsened patients, by 0.66 in elderly in persistently pre-frail patients and by 0.80 in persistently frail patients.

## 3.6 Linear regression analysis of the relationship between frailty transitions and 2-year HRQoL

Table 4 show that when the above confounders were adjusted, the regression coefficient of HRQoL within 2 years after enrollment was -1.23 (95% *CI*: -2.12 to -0.35) for frailty-improved patients, -3.67 (95% *CI*: -4.71 to -2.62) for frailty-worsened patients, -4.95 (95% *CI*: -5.96 to -3.94) for persistently pre-frail patients and -9.76 (95% *CI*: -11.60 to -7.93) for persistently frail patients. Compared with the persistently non-frail patients, HRQoL decreased by 1.23 over 2 years in frailty-improved patients, by 3.67 in frailty-worsened patients, by 4.95 in persistently pre-frail patients and by 1.23 in persistently frail patients.

## 4 Discussion

In this nationally representative sample of elderly inpatients in China, we found a prevalence of frailty and pre-frailty of 40.70% and 43.04%, respectively. Moreover, the results of this study show that the prevalence of frailty is higher in women than in men and increases with age. These observations provide further evidence that a substantial proportion of elderly hospitalized patients are frail. The internationally reported prevalence of frailty among hospitalized elderly patients ranges from 17.9% to 40% [14]; the prevalence of frailty in elderly inpatients defined using the FRAIL scale in this study was comparable to that reported in other studies [36]. It should be noted that differences in the prevalence of frailty depend on the frailty assessment tool used and the population being assessed [37]. The main implication of our study and previous studies is that frailty is common among elderly inpatients and must be taken into account. The high prevalence of pre-frailty in hospitalized patients should not be overlooked, with studies reporting prevalence rates ranging from 9% to 58.3% [14]. Frailty is a dynamic process that can improve or worsen. It is common for older adults to progress from pre-frailty to frailty during hospitalization [38, 39]. Pre-frailty is a state in which multiple physiological systems begin to decline, but reserves remain sufficient to withstand most stresses [40]. Studies have shown that older adults with pre-frailty had a lower risk of experiencing falls [41], hospital readmission and death than older adults with frailty [4].

Frail individuals often have reduced physical activity, leading to muscle atrophy. However, improvements in frailty status can lead to increased physical activity, which in turn reduces the risks of mortality, falls, and hospital readmission [42]. Therefore, identifying frailty at an early stage and providing timely and effective interventions for frail elderly adults might be beneficial in minimizing the risk of adverse outcomes in this population [43].

Frailty in the elderly is a dynamic process, with frailty status transitioning over time. At the time of the three-month follow-up, more than 1/2 patients maintained the same frailty status, of which 6.25% remained frail, 18.20% remained pre-frail and 30.09% remained non-frail. Another 15.90% worsened and 29.55% improved. Our study found that more participants improved rather than worsened frailty status, while Kojima et al. showed that progression of frailty condition was more common than improvement [44]. This inconsistency may be due to the fact that the subjects of this study were hospitalized elderly patients, some of whom had a decline in their physical conditions after surgery or acute illness, and the frailty syndrome ensued or worsened. But after 3 months of recuperation, the frailty condition improved with the recovery of the physical improvement. Previous studies have reported that polypharmacy is prevalent among elderly inpatients and is a risk factor for worsening frailty and mortality [45]. Our study results also indicate that frailty was more prevalent among patients with baseline polypharmacy (S1 Table). Controlling and optimizing medication use in older adults may help mitigate the progression of frailty. In addition, the rate of change in frailty was associated with both gender and age. Trevisan et al. found that women were more likely to change frailty status than men, rather than remaining the same [46]. And a study suggested that as people age, the likelihood of becoming frail or worsening frailty increased [44].

Previous studies shown that baseline frailty is a risk factor for adverse outcomes such as death and falls [4, 47, 48], but there are few studies on the association between longitudinal changes in frailty and adverse outcomes. Our study found that frailty-improved patients had a 1.32 times higher risk of death than persistently non-frail patients. This may be due to the fact that the frailty status of the elderly inpatients improved after treatment and recovery, but their physical health conditions were relatively unstable. The risk of death for frailty-worsened patients (*HR* = 2.43) was higher than for persistently frail patients (*HR* = 2.44), which was inconsistent with the results of the cohort study in Taiwan Province [49]. The reasons may be the short follow-up period of this study, and for the elderly inpatients, gradually worse frailty reflected a progressively poor physical health and was an important predictor of death.

The 2-year follow-up results revealed a significant association between frailty transition patterns and falls in elderly inpatients, indicating that improvement in frailty, worsening of frailty, persistent pre-frailty and persistent frailty are risk factors for falls in elderly inpatients. Moreover, frailty-worsened patients and persistently pre-frail patients had an increased number of falls relative to persistently frail non-patients. Falls result in serious health consequences among older people such as limited activity, disability, and death [50]. Falls have become an important issue in the field of public health, and falls are the leading cause of death from injury among adults aged ≥65 years in China [51]. Falls occur in 20.7% of elderly adults, and studies have reported that 5% to 15% of older individuals who experience a fall can also experience brain injury, fracture, and soft tissue injury [52]. Falls can also have adverse effects on mental health [53], such as reduced activity owing to a fear of falling, leading to reduced physical function, social maladjustment, and an increased risk of falling [54]. The occurrence of frailty and falls is related to decreased bone mass, decreased muscle mass, and sarcopenia [55], and frailty affects mobility prior to the occurrence of adverse clinical outcomes such as falls. Therefore, the early stages of frailty are the best time to perform interventions aimed at preventing mobility impairment, thereby avoiding falls in the future. The relationship between frailty and falls is bidirectional. In elderly people, falls can lead to frailty and frailty can lead to falls.

In this study, we found a significant association between frailty transitions and increased risk of rehospitalization. Moreover, frailty-improved, frailty-worsened, persistently pre-frail and persistently frail elderly inpatients had a higher risk of rehospitalization and an increased total number of days of rehospitalization relative to persistently non-frail elderly inpatients. Plausible explanations for this association included the fact that a frail condition in elderly patients affects functional recovery of the body, leading to a poorer prognosis for other diseases and a resulting increase in the use of medical services [43]. Additionally, frail elderly people are more vulnerable to stress events and more likely to be overburdened by adverse health events such as falls, resulting in an increased risk of hospital readmission [36].

Studies have reported that 16%–35% of frail individuals also have depression [56], and patients with frailty and depression are highly overlapping populations [57]. The congruence of the diagnostic criteria for these two diseases with respect to some key measurement items may explain this overlap, such as fatigue and weight loss [58]. Furthermore, Mezuk et al. revealed a potential bidirectional association between depressive symptoms and frailty [59]. Depression in later life increases the risk of developing frailty [60]. Depressed individuals may be more likely to be frail owing to reduced sleep duration and quality, lack of physical activity, poor nutritional intake, and poor adherence to medications [61]. Similarly, those who are frail may be at greater risk of developing depressive symptoms owing to emotional distress resulting from continued physical decline and diminishing social contact [62]. The two diseases share common pathophysiological mechanisms, including subclinical cerebrovascular disease, chronic inflammation, and neuroendocrine dysregulation [58]. As well, a significant interaction existed between cognitive impairment and frailty [63]. Lee reported a 92% higher risk of death in participants with frailty and cognitive impairment compared with elderly inpatients who did not have frailty or cognitive impairment [64].

The strengths of this study are that elderly inpatients from six general hospitals across China were included in this large-scale, nationally representative study. Additionally, since frailty is a dynamically changing process, we constructed the frailty transition as a variable to replace baseline decline to explore its relationship with adverse outcomes. As the subjects of this study were elderly inpatients from multiple departments of general hospitals, some patients may exhibit symptoms of frailty syndrome due to certain acute illnesses or poor physical condition after surgery, but after several months of treatment and recovery, their physical condition improved and the frailty symptoms also subsequently improved. At this time, if baseline frailty were still used to assess its association with adverse outcomes, the association would be overestimated [43].

Our study also has several limitations. Firstly, although we found an association between frailty transitions and an increased risk of adverse health outcomes in this prospective cohort study, it should be noted that adverse outcomes such as falls, functional decline, and death are common during hospitalization and may result from the health problems that led to hospitalization or the consequences of hospitalization. Secondly, information bias caused by recall bias and self-reported symptoms could not be completely avoided when using patient-reported information about falls and HRQoL over the past year, especially for the 19.81% of patients with cognitive impairment. Thirdly, the FRAIL scale was adopted to assess frailty in this study, and the findings cannot be well extrapolated to studies using other frailty assessment tools. Fourthly, the severity of patients' conditions differed between tertiary hospitals and other medical institutions, and there was selection bias in the study population.

## 5 Conclusion

We observed that frailty-improved, frailty-worsened, persistently pre-frail and persistently frail elderly inpatients are at higher risk of subsequent mortality, falls, rehospitalization, and reduced HRQoL than persistently non-frail elderly inpatients. Our findings suggested that frailty is a dynamic process and can be improved, so frailty assessment and risk stratification are essential in patient and clinician planning, as well as health care resource utilization. Attention to the early identification of frailty and development of an evidence-base to inform best practices for managing frail older individuals may help to improve outcomes in this population.

## Supporting information

**S1 Table. Prevalence of frailty across demographic groups (n [%]).**
(DOCX)

## Acknowledgments

The authors thank the older inpatients, nurses, clinicians, and management staff of the research hospitals that were involved in the study. We thank Analisa Avila, MPH, ELS, of Liwen Bianji (Edanz) (www.liwenbianji.cn), for editing the language of a draft of this manuscript.

## Author Contributions

**Investigation:** Jiaqi Ding, Xinjuan Wu, Xianxiu Wen, Hui Wang, Dongmei Lv, Shengxiu Zhao.

**Methodology:** Jing Jiao, Tao Xu.

**Writing – original draft:** Miao Yu.

**Writing – review & editing:** Jingfen Jin, Jing Jiao, Tao Xu.

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
