## [Decision Letter · Decision Letter 0]

9 Sep 2024

PONE-D-24-15744Indication of frailty transitions on 2-year adverse health outcomes among older Chinese inpatients: insight from a multicenter prospective cohort studyPLOS ONE

Dear Dr. Xu,

Thank you for submitting your manuscript to PLOS ONE. After careful consideration, we feel that it has merit but does not fully meet PLOS ONE’s publication criteria as it currently stands. Therefore, we invite you to submit a revised version of the manuscript that addresses the points raised during the review process.

We look forward to receiving your revised manuscript.

Kind regards,

Marina De Rui, MD PhD

Academic Editor

PLOS ONE

Journal Requirements:

   "NO"

5. We note that your Data Availability Statement is currently as follows: All relevant data are within the manuscript and its Supporting Information files

Reviewers' comments:

Reviewer's Responses to Questions

**Comments to the Author**

1. Is the manuscript technically sound, and do the data support the conclusions?

Reviewer #1: Yes

Reviewer #2: Yes

2. Has the statistical analysis been performed appropriately and rigorously? 

Reviewer #1: Yes

Reviewer #2: I Don't Know

3. Have the authors made all data underlying the findings in their manuscript fully available?

Reviewer #1: Yes

Reviewer #2: Yes

4. Is the manuscript presented in an intelligible fashion and written in standard English?

Reviewer #1: Yes

Reviewer #2: Yes

5. Review Comments to the Author

Reviewer #1: In this paper, You and colleagues estimated the association between changes in frail status and adverse outcomes among Chinese elderly inpatients. In this longitudinal multicenter cohort study the authors assessed frailty using the FRAIL (Fatigue, Resistance, Ambulation, Illness, and Loss of weight) scale and divided participants into 3 groups (non-frailty, pre-frailty, frailty). Frailty-improved, worsened, persistently pre-frail, and frail inpatients faced higher risks of mortality, falls, rehospitalization, reduced quality of life compared to consistently non-frail inpatients.

The work is based on a solid sample selection, it is rigorously designed and clearly written.

I have a few minor concerns.

Background

The background section could benefit from a more detailed review of the literature, and the addiction of a few more citations (e.g. lines 69, 72).

Line 75: I think the authors should clarify what they mean with “acceleration of population aging”.

Line 91: the authors could add citations of studies that employed frailty in relation to intermediate and long-term outcomes. I would also remove the sentence “In contrast, the present study investigated the influence of frailty transitions on these outcomes.” or move it to the end of the Background section were the aim of the present study are described.

Line 111: The authors could further explain the concept of HRQoL.

Materials and methods

Line 153: the authors could clarify what they mean with “history of surgery” (e.g. during lifetime or as the reason of patient’s admission to hospital). It would be also useful to add a variable for the reason of admission (e.g. macro-groups such as “surgery” or “acute medical condition”), that could also be used to evaluate long-term outcomes.

Results

In general I would suggest the authors to keep two significant figures in expressing results to add clarity and keep the text lighter.

I would also suggest to provide a flowchart of the patient selection and follow up (with numbers of people excluded or lost at follow up).

Line 217: I think that the authors refer to Supplementary Table 1.

Discussion

Line 324: “The prevalence of frailty is higher in women than in men and increases with age”: it is not clear whether the authors refer to a finding of theirs or a datum from literature; in both cases a citation would demonstrate comparison with data already available in the literature.

Line 364: “Furthermore, patients with one or two components of frailty phenotype were also considered non-frail when constructing the variable for frailty transition, which may also increase the risk of death in the frailty-improved group.”: please rephrase because it is not clear what you mean here.

Line 373: the authors should try to explain why also an improvement in frailty is associated with falls.

Conclusion

Line 445: “Our findings suggested that frailty is a dynamic process and can be improved”: as stated before, I would recommend to give a further explanation of what happens in frailty-improved individuals before making such a statement.

Reviewer #2: 1. Very important topic for Geriatric population on frailty

2. Classification of frailty and logically presented with data supporting the transitions.

3. Study has several limitations due to biases and disproportionate collection of data from different levels of hospital care.

4. Out of the patients, can you provide data on the diagnosis, level of care of the hospitalized patients to assess the differences on frailty based on the severity of the illness.

5. Does fall risk increase in patients with previous fall risk or falls? Any additional data on these patients. '

6. Were there any other data including polypharmacy resulting in frailty?

7. Line 430- word may is repeated.

8. Need clear exclusion criteria and the level of training for the nurses for conducting the study elaborated.

9. Overall important study for geriatric population and frailty.

6. PLOS authors have the option to publish the peer review history of their article (what does this mean?). If published, this will include your full peer review and any attached files.

Reviewer #1: No

Reviewer #2: No

---

## [Author Response · Author response to Decision Letter 0]

9 Oct 2024

Dear editor:

Thank you for your work. We have received the comments of all the reviewers. The following are the comments made by the reviewers and all authors responses.

Reviewer #1: 

Background

The background section could benefit from a more detailed review of the literature, and the addiction of a few more citations (e.g. lines 69, 72).

Thank you for your suggestion. I have added the following content to the first paragraph: China's population is already aging, and it will continue to age rapidly in the coming decades. In 2021, 14.2% of China's population was aged 65 and over, signifying that the proportion of elderly individuals aged 65 and above in China has doubled within 21 years, increasing from 7% to 14%

1. Chen X, Giles J, Yao Y, Yip W, Meng Q, Berkman L, et al. The path to healthy ageing in China: a Peking University-Lancet Commission. Lancet. 2022;400(10367):1967-2006. Epub 20221121. doi: 10.1016/s0140-6736(22)01546-x. PubMed PMID: 36423650; PubMed Central PMCID: PMCPMC9801271.

Line 75: I think the authors should clarify what they mean with “acceleration of population aging”.

We have replaced the sentence with: “With the rapid progression of population aging, frailty has been increasingly recognized for its significant impact on healthcare systems worldwide.”

Line 91: the authors could add citations of studies that employed frailty in relation to intermediate and long-term outcomes. I would also remove the sentence “In contrast, the present study investigated the influence of frailty transitions on these outcomes.” or move it to the end of the Background section were the aim of the present study are described.

Thank you for your suggestion. I have added the relevant references and removed the sentence "In contrast, the present study investigated the influence of frailty transitions on these outcomes" as per your recommendation.

17. Chi J, Chen F, Zhang J, Niu X, Tao H, Ruan H, et al. Frailty is associated with 90-day unplanned readmissions and death in patients with heart failure: A longitudinal study in China. Heart Lung. 2022;53:25-31. Epub 20220201. doi: 10.1016/j.hrtlng.2022.01.007. PubMed PMID: 35121488.

18. Shi J, Shi B, Tao YK, Meng L, Zhou ZY, Chen SQ, et al. [Relationship between frailty status and risk of death in the elderly based on frailty index analysis]. Zhonghua Liu Xing Bing Xue Za Zhi. 2020;41(11):1824-30. doi: 10.3760/cma.j.cn112338-20200506-00691. PubMed PMID: 33297646.

Line 111: The authors could further explain the concept of HRQoL.

Thank you for your suggestion. I have now explained the concept of HRQoL in the article as follows, with references included: This study aims to use the Health-Related Quality of Life (HRQoL) metric to assess patients' health and living conditions. HRQoL is defined by the World Health Organization as "an individual's perception of their position in life, in the context of the culture and value systems in which they live, and in relation to their goals, expectations, standards, and concerns."

20. The World Health Organization Quality of Life assessment (WHOQOL): position paper from the World Health Organization. Soc Sci Med. 1995;41(10):1403-9. doi: 10.1016/0277-9536(95)00112-k. PubMed PMID: 8560308.

Materials and methods

Line 153: the authors could clarify what they mean with “history of surgery” (e.g. during lifetime or as the reason of patient’s admission to hospital). It would be also useful to add a variable for the reason of admission (e.g. macro-groups such as “surgery” or “acute medical condition”), that could also be used to evaluate long-term outcomes.

I have clearly defined the surgical history in the manuscript: history of surgery (defined as having undergone surgery at any point in life).Thank you for your constructive suggestion. However, we have not yet compiled data on admission diagnoses. Nevertheless, our study included variables such as history of surgery, falls within the past year, bedridden status for ≥ 4 weeks, use of multiple medications, nutritional status, presence of depression, cognitive function, urination and defecation patterns, tumor presence, Activities of Daily Living (ADL), and Instrumental Activities of Daily Living (IADL) to assess the severity of the patients' conditions and baseline health differences among frail patients. These variables can also be used to evaluate long-term outcomes.

Results

In general I would suggest the authors to keep two significant figures in expressing results to add clarity and keep the text lighter.

Thank you for your suggestion. We have adjusted the results in the text and tables to two significant figures for added clarity and to keep the presentation concise.

I would also suggest to provide a flowchart of the patient selection and follow up (with numbers of people excluded or lost at follow up).

Thank you for your valuable suggestions. I have added the flowchart of patient selection and follow-up (including the number of individuals excluded or lost during follow-up) as per your request.

Line 217: I think that the authors refer to Supplementary Table 1.

Thank you for your careful review. What I intended to express is: Supplementary Table 1, and I have already corrected it in the manuscript.

Discussion

Line 324: “The prevalence of frailty is higher in women than in men and increases with age”: it is not clear whether the authors refer to a finding of theirs or a datum from literature; in both cases a citation would demonstrate comparison with data already available in the literature.

Thank you for your suggestion. We have revised the sentence to: "Moreover, the results of this study show that the prevalence of frailty is higher in women than in men and increases with age" to eliminate ambiguity.

Line 364: “Furthermore, patients with one or two components of frailty phenotype were also considered non-frail when constructing the variable for frailty transition, which may also increase the risk of death in the frailty-improved group.”: please rephrase because it is not clear what you mean here.

This sentence means that, when defining frailty transitions based on the FRAIL scale, some patients who met one or two criteria—indicating they were not in perfect health—were still classified as non-frail, which could increase their risk of death. However, as this sentence adds little value and may cause confusion, I will remove it from the manuscript.

Line 373: the authors should try to explain why also an improvement in frailty is associated with falls.

Since we used the "remaining non-frail" group as the reference, the individuals who improved from frailty might have had poor physical conditions previously.

Conclusion

Line 445: “Our findings suggested that frailty is a dynamic process and can be improved”: as stated before, I would recommend to give a further explanation of what happens in frailty-improved individuals before making such a statement.

Thank you for your valuable suggestion. We have added the following content: Frail individuals often have reduced physical activity, leading to muscle atrophy. However, improvements in frailty status can lead to increased physical activity, which in turn reduces the risks of mortality, falls, and hospital readmission[42].

42. Valenzuela PL, Saco-Ledo G, Morales JS, Gallardo-Gómez D, Morales-Palomo F, López-Ortiz S, et al. Effects of physical exercise on physical function in older adults in residential care: a systematic review and network meta-analysis of randomised controlled trials. Lancet Healthy Longev. 2023;4(6):e247-e56. Epub 20230511. doi: 10.1016/s2666-7568(23)00057-0. PubMed PMID: 37182530.

Reviewer #2: 

1.Very important topic for Geriatric population on frailty

Thank you for recognizing the importance of our work, particularly on the topic of frailty in the geriatric population. We fully understand the significance of frailty as a critical issue in elderly healthcare, and its study and clinical interventions are vital for improving the quality of life in older adults. Your positive feedback reinforces the value of our research and motivates us to continue exploring this area, with the goal of aiding healthcare professionals in identifying high-risk individuals and implementing appropriate interventions. Once again, we greatly appreciate your support and valuable comments.

2. Classification of frailty and logically presented with data supporting the transitions.

Thank you for your positive feedback on our classification of frailty and the clear presentation of the transitions. We are committed to accurately classifying frailty states with data-supported approaches and demonstrating the transition processes to provide more practical guidance for clinical practice. Your recognition is highly encouraging and will continue to drive us forward in further exploring this important research area.

3. Study has several limitations due to biases and disproportionate collection of data from different levels of hospital care.

First, we acknowledge that there may be bias in our study as the data were collected from tertiary hospitals, which we have mentioned in the limitations section. If possible, we will aim to collect data from hospitals at different levels in future research. However, since our data come from tertiary hospitals, the diagnosis of diseases is more accurate, patient adherence is better, and the loss-to-follow-up rate is relatively low.

4. Out of the patients, can you provide data on the diagnosis, level of care of the hospitalized patients to assess the differences on frailty based on the severity of the illness.

Thank you for your suggestion. However, our study included variables such as history of surgery, falls within the past year, bedridden status for ≥ 4 weeks, use of multiple medications, nutritional status, presence of depression, cognitive function, urination and defecation patterns, tumor presence, Activities of Daily Living (ADL), and Instrumental Activities of Daily Living (IADL) to assess the severity of the patients' illnesses and the baseline health differences among frail patients.

5. Does fall risk increase in patients with previous fall risk or falls? Any additional data on these patients. 

In this study, "previous fall risk or history of falls" was included as a covariate. As shown in Supplementary Table 1, patients with a previous fall risk or history of falls had a higher frailty risk. In Table 2, it is shown that persistent frailty leads to a higher risk of falls over a two-year period. Therefore, it can be inferred that patients with previous fall risk or a history of falls may have an increased fall risk. However, this study did not directly analyze the association between these two variables.

6. Were there any other data including polypharmacy resulting in frailty?

Thank you for raising this important question. We have added the following content to the manuscript: 'Previous studies have reported that polypharmacy is prevalent among elderly inpatients and is a risk factor for worsening frailty and mortality[45]. Our study results also indicate that frailty was more prevalent among patients with baseline polypharmacy (Supplementary Table 1). Controlling and optimizing medication use in older adults may help mitigate the progression of frailty.' A reference has been included.

45. Liu X, Zhao R, Zhou X, Yu M, Zhang X, Wen X, et al. Association between polypharmacy and 2-year outcomes among Chinese older inpatients: a multi-center cohort study. BMC Geriatr. 2024;24(1):748. Epub 20240909. doi: 10.1186/s12877-024-05340-3. PubMed PMID: 39251936; PubMed Central PMCID: PMCPMC11382416.

7. Line 430- word may is repeated.

Thank you for your careful review. I have removed the repeated word "may".

8. Need clear exclusion criteria and the level of training for the nurses for conducting the study elaborated.

Thank you for your feedback. In response to your concern, we have elaborated on the exclusion criteria and the training provided to the nurses involved in the study as follows:

Exclusion criteria: ① Patients with persistent consciousness disorders or communication disorders that hinder effective communication, and whose caregivers are unable to provide accurate information; ② Participants lost to follow-up during the 2-year follow-up period after enrollment.

Training for nurses: We provided standardized training on study procedures, data collection, communication, and the EDC system. Nurses were also trained in data accuracy and quality control, with periodic refresher sessions. We hope this provides clarity and addresses your concerns.

9. Overall important study for geriatric population and frailty.

Thank you once again for your positive feedback on our study. Your recognition greatly encourages us and motivates us to continue our exploration in the field of geriatric population and frailty research. We will keep striving to ensure that our research contributes significantly to clinical practice and the health management of older adults.

Here is the reference I have added:

1. Chen X, Giles J, Yao Y, Yip W, Meng Q, Berkman L, et al. The path to healthy ageing in China: a Peking University-Lancet Commission. Lancet. 2022;400(10367):1967-2006. Epub 20221121. doi: 10.1016/s0140-6736(22)01546-x. PubMed PMID: 36423650; PubMed Central PMCID: PMCPMC9801271.

17. Chi J, Chen F, Zhang J, Niu X, Tao H, Ruan H, et al. Frailty is associated with 90-day unplanned readmissions and death in patients with heart failure: A longitudinal study in China. Heart Lung. 2022;53:25-31. Epub 20220201. doi: 10.1016/j.hrtlng.2022.01.007. PubMed PMID: 35121488.

18. Shi J, Shi B, Tao YK, Meng L, Zhou ZY, Chen SQ, et al. [Relationship between frailty status and risk of death in the elderly based on frailty index analysis]. Zhonghua Liu Xing Bing Xue Za Zhi. 2020;41(11):1824-30. doi: 10.3760/cma.j.cn112338-20200506-00691. PubMed PMID: 33297646.

20. The World Health Organization Quality of Life assessment (WHOQOL): position paper from the World Health Organization. Soc Sci Med. 1995;41(10):1403-9. doi: 10.1016/0277-9536(95)00112-k. PubMed PMID: 8560308.

42. Valenzuela PL, Saco-Ledo G, Morales JS, Gallardo-Gómez D, Morales-Palomo F, López-Ortiz S, et al. Effects of physical exercise on physical function in older adults in residential care: a systematic review and network meta-analysis of randomised controlled trials. Lancet Healthy Longev. 2023;4(6):e247-e56. Epub 20230511. doi: 10.1016/s2666-7568(23)00057-0. PubMed PMID: 37182530.

45. Liu X, Zhao R, Zhou X, Yu M, Zhang X, Wen X, et al. Association between polypharmacy and 2-year outcomes among Chinese older inpatients: a multi-center cohort study. BMC Geriatr. 2024;24(1):748. Epub 20240909. doi: 10.1186/s12877-024-05340-3. PubMed PMID: 39251936; PubMed Central PMCID: PMCPMC11382416.

Thank you for your kind attention and looking forwards to your favorable reply.

Yours sincerely,

Tao Xu

Department of epidemiology and statistics

Institute of Basic Medical Sciences, Chinese Academy of Medical Sciences & School of Basic Medicine, Peking Union Medical College

5, Dong dan san tiao

Beijing 100005

China

Tel: 86 10 69156408

E-mail: 

xutaosd@126.com

---

## [Editor Report · Decision Letter 1]

31 Oct 2024

Indication of frailty transitions on 2-year adverse health outcomes among older Chinese inpatients: insight from a multicenter prospective cohort study

PONE-D-24-15744R1

Dear Dr. Xu,

We’re pleased to inform you that your manuscript has been judged scientifically suitable for publication and will be formally accepted for publication once it meets all outstanding technical requirements.

Kind regards,

Marina De Rui, MD PhD

Academic Editor

PLOS ONE
---

## [Editor Report · Acceptance letter]

6 Nov 2024

PONE-D-24-15744R1 

PLOS ONE

Dear Dr. Xu, 

I'm pleased to inform you that your manuscript has been deemed suitable for publication in PLOS ONE. Congratulations! Your manuscript is now being handed over to our production team.

Kind regards, 

on behalf of

Dr. Marina De Rui 

Academic Editor

PLOS ONE